# Highly Thermal Conductive and Electrical Insulating Epoxy Composites with a Three-Dimensional Filler Network by Sintering Silver Nanowires on Aluminum Nitride Surface

**DOI:** 10.3390/polym13050694

**Published:** 2021-02-25

**Authors:** Wondu Lee, Jooheon Kim

**Affiliations:** 1School of Chemical Engineering & Materials Science, Chung-Ang University, Seoul 156-756, Korea; sems212@naver.com; 2Department of Intelligent Energy and Industry, Chung-Ang University, 84 Heukseok-ro, Dongjak-gu, Seoul 156-756, Korea

**Keywords:** polymer-matrix composites (PMCs), thermal conductivity, 3D network, sintering

## Abstract

In this study, a new fabrication technique for three-dimensional (3D) filler networks was employed for the first time to prepare thermally conductive composites. A silver nanowire (AgNW)– aluminum nitride (AlN) (AA) filler was produced by a polyol method and hot-pressed in mold to connect the adjacent fillers by sintering AgNWs on the AlN surface. The sintered AA filler formed a 3D network, which was subsequently impregnated with epoxy (EP) resin. The fabricated EP/AA 3D network composite exhibited a perpendicular direction thermal conductivity of 4.49 W m^−1^ K^−1^ at a filler content of 400 mg (49.86 vol.%) representing an enhancement of 1973% with respect to the thermal conductivity of neat EP (0.22 W m^−1^ K^−1^). Moreover, the EP/AA decreased the operating temperature of the central processing unit (CPU) from 86.2 to 64.6 °C as a thermal interface material (TIM). The thermal stability was enhanced by 27.28% (99 °C) and the composites showed insulating after EP infiltration owing to the good insulation properties of AlN and EP. Therefore, these fascinating thermal and insulating performances have a great potential for next generation heat management application.

## 1. Introduction

In recent decades, the rapid advancement of automotive electrification technologies has considerably increased the demand for high-performance electronic devices [1,2]. Owing to the high degrees of miniaturization [3] and the integration [4] of these devices, a large amount of heat is accumulated near the drivetrain, which strongly affects the durability of driving components and may cause vehicle malfunction. Therefore, the proper heat management for electronic devices is considered a critical issue [5,6]. To dissipate the heat, the polymer composites are widely introduced in device packaging as a thermal interface materials (TIMs) [7,8]. Despite the low weight, high processability and low cost of polymeric materials such as epoxy (EP) [9], polydimethylsiloxane [10], cellulose nanofibers [11] and polyvinyl alcohol [12], their intrinsically low thermal conductivities (0.2–0.5 W m^−1^ K^−1^) [13] constitute a bottleneck in the fabrication of thermally conductive composites. To address this problem, highly thermally conductive materials, such as ceramic fillers composed of boron nitride (BN) [14] and aluminum nitride (AlN) [15], carbon-based graphene and carbon nanotubes [16], and metallic fillers comprising silver nanowires (AgNWs) [17] and copper (Cu) [18], are integrated within polymer materials to increase their thermal conductivities. AlN is a popular ceramic filler that is used to fabricate highly thermally conductive and insulating composites because of its intrinsically low dielectric constant (8.8), excellent electrical resistance (> 10^14^ Ω·cm), high thermal conductivity (100–300 W m^−1^ K^−1^), and low thermal expansion coefficient (4 × 10^−6^ K^−1^) [19]. Ag has received much attention from researchers, owing to its high electrical conductivity (6.30 × 10^7^ S m^−1^) and thermal conductivity (427 W m^−1^ K^−1^) as compared with those of other metals [20]. In addition, AgNWs and Ag nanoparticles have been used in various fields, including catalysis [21], microelectronics [22] and thermally conductive composites [23].

However, the intrinsic properties of materials can be enhanced only to a certain extent. The properties of the composites are affected by the utilized fabrication methods (such as hot pressing [24], plastic injection [25], and vacuum filtration [26]). In particular, the three-dimensional (3D) filler networks incorporated into polymer matrices via freeze-casting [27], chemical vapor deposition (CVD) [28] and foaming induced by mixing [29] have been recently investigated by various groups. Zhou et al. fabricated 3D graphene fillers (3DGFs)/EP composites by a CVD method using a Ni template. The obtained composites exhibited a thermal conductivity of 0.52 W m^−1^ K^−1^ at a filler content of 0.14 vol.%, which was 1.74 times higher than that of neat EP [30]. Wei et al. fabricated anisotropic EP/aluminum nitride honeycomb composites via a freeze-casting method. The obtained materials possessed in-plane and through-plane thermal conductivities of 9.48 and 4.45 W m^−1^ K^−1^ at a filler content of 47.26 vol.%, respectively [31]. Xu et al. prepared 3D BN foam-filled EP composites by mixing, decomposition, and infiltration techniques, which exhibited a high thermal conductivity of 6.11 W m^−1^ K^−1^ at a filler content of 59.43 vol.% [32].

In this study, AlN particles with the wurtzite structure and AgNWs prepared by the reduction of silver nitrate (AgNO_3_) by a polyol method [33,34] was used to fabricate a highly thermally conductive 3D filler network in an EP matrix. First, a chestnut bur-like filler (marked AA) was obtained by growing AgNWs from AgNO_3_ seeds attached to the AlN particle surface. The fillers were then connected into a 3D filler network by sintering via hot pressing inside a mold. The pores in the resulting 3D filler network were filled by the EP resin and cured. The obtained EP/AA 3D network composites exhibited excellent thermal and insulation properties, suggesting that the developed composites could be potentially used as TIMs to improve the heat management of automotive electronics and satisfy the existing demand for thermally conductive and well-insulated composite materials.

## 2. Experimental

### 2.1. Materials

AlN powder, with an average particle diameter of 4 μm and a density of 3.26 g/cm^3^, was supplied from Alfa Aesar (Haverhill, MA, USA). AgNO_3_ (molecular weight (MW) = 169.87, 99.8%), ethylene glycol (EG) (99.0%), polyvinylpyrrolidone (PVP, MW = 40,000, K30), and sodium chloride standard solution (NaCl, 0.1 M) were obtained from Daejung Chemical Co. (Seoul, Korea). Diglycidyl ether of bisphenol-A (DGEBA, EP equivalent weight ¼ 186.8 g/eq) was supplied by Kukdo Chemical Co. (Seoul, Korea). 4,4’-Diaminodiphenylmethane (DDM, 97.0%) purchased from Sigma-Aldrich (St. Louise, MO, USA) was used as the curing agent. All reagents and solvents were used as received without further purification.

### 2.2. Preparation of the AgNW- Deposited AlN Filler

The polyol method was used to prepare the AgNW- deposited AlN filler. First, AlN and AgNO_3_ (1 g) were mixed at weight ratios of 1:2, 1:1, 2:1, and 3:1 and stirred gently in EG (40 mL) at room temperature for 2 h (solution 1). PVP (2.64 g) and NaCl (2 mL) were mixed and stirred in EG (50 mL) at 90 °C until all PVP species were dissolved completely (solution 2). After the color of solution 1 became yellow, it was poured into solution 2 and agitated at 90 °C until the color of the mixed solution turned orange, followed by autoclaving at 160 °C for 8 h. Subsequently, the solution color changed to gray because of the reduction of Ag species, which were diluted with ethanol and centrifuged at 10,000× *g* rpm for 20 min to remove the residual PVP. The final AA filler product was obtained by filtration (nylon membranes, pore size: 0.45 µm, diameter: 47 mm) using ethanol as solvent and drying in an oven at 70 °C overnight.

### 2.3. Preparation of the 3D Network and EP Composites

To achieve excellent heat management performance of the produced composites, a 3D network comprising AlN and AgNWs was fabricated by the polyol method followed by hot pressing. The 3D network was prepared from the AA filler by hot pressing. The AA filler was poured into a mold and was hot-pressed at a temperature of 310 °C and pressure of 14 MPa for 30 s to sinter the AgNWs on the AlN surface. A disc-shaped 3D filler network was obtained after hot pressing. EP/AA 3D network composites were produced by the incorporation of EP resin into the 3D filler network. For this purpose, DGEBA EP resin (0.5 g) and DDM (0.2 g) curing agent were mixed and stirred slowly at a temperature of 110 °C until their complete dissolution. The prepared mixture was poured into a polytetrafluoroethylene (PTFE) disc mold containing the 3D filler network, followed by the bubble removal procedure conducted in a vacuum oven at 70 °C for 30 min. The final composites were produced by curing in a box furnace at 180 °C for 30 min.

### 2.4. Characterization

The filler morphology and composite cross-sections were analyzed by field–emission transmission electron microscopy (FE–TEM, electron high tension voltage: 200 kV, JEOL Ltd., JEM-F200, Tokyo, Japan), field–emission scanning electron microscopy (FE–SEM, electron high tension voltage: 5 kV, Carl Zeiss, SIGMA, Oberkochen, Germany), and energy-dispersive X-ray spectroscopy (EDS) combined with FE–SEM. X-ray diffraction (XRD, Bruker-AXS, D8-Advance) patterns were examined to analyze the crystalline structures of the AA filler and 3D filler network (0.2° s^−1^ scan rate and Cu Kα radiation (0.154056 nm) in a 2θ range of 10–80°). The glass transition temperature (T_g_) of AgNWs was measured by differential scanning calorimetry (DSC, Perkin-Elmer Co., DSC-7, Waltham, MA, USA) in a temperature range from 20 to 400 °C. A laser flash analysis (LFA, Netzsch Instruments Co., NanoFlash LFA 467, Selb, Germany) was used to examine the thermal diffusivities and specific heat capacity of the composites with the laser flash pulse directed through a disc (diameter: 12.7 mm) specimen at room-temperature. Thermal conductivities of the fabricated EP/AA 3D network composites were calculated by the equation *K* = *α* × *ρ* × *Cp*, where *K* is thermal conductivity (W m^−1^ K^−1^), *α* is thermal diffusivity (mm^2^ s^−1^), *ρ* is bulk density (g cm^−3^), and *Cp* is room temperature heat capacity (J g^−1^ K^−1^). The thermal stabilities and thermal degradations of the EP/AA 3D network composites were analyzed by thermogravimetric analysis (TGA, TGA-2050, TA Instruments, New Castle, DE, USA), which was performed by heating them to 800 °C at a rate of 10 °C min^−1^ under nitrogen atmosphere. A four-point probe method (Keithley, 2400 Source Meter, Cleveland, OH, USA) was used to determine the electrical conductivities of the composites, and their thicknesses were measured using a digital micrometer.

## 3. Results and Discussion

### 3.1. Composite Preparation Process and Filler Morphology

A schematic illustration of the fabrication process is presented in Figure 1. First, AgNO_3_ seed and AlN powder were gently stirred in EG to deposit AgNO_3_ seeds onto the AlN surface via van der Waals forces [35]. Subsequently, AgNWs were reduced and grown on these AgNO_3_ seeds by the polyol method. The produced AA fillers were poured into a mold and hot-pressed at 310 °C corresponding the temperature above T_g_ of the AgNWs. After hot pressing, a disc-shaped 3D network was successfully prepared by sintering AgNWs on the AlN surface, and the EP/AA composites were fabricated via EP impregnation.

The morphology of the AA filler determined by FE–SEM and EDS is illustrated in Figure 2. Here, the particles of raw AlN and wire-shaped AgNW are presented in Figure 2a,b, respectively, while the AgNWs deposited on the AlN surface are depicted in Figure 2c; the corresponding FE–SEM EDS image is shown in Figure 2d–f. The obtained EDS distribution maps of Al, N, and Ag in the AA filler indicate that AgNWs were grown successfully on the surface of AlN particles from AgNO_3_ seeds. Moreover, the AA fillers of various weight ratios between the AlN and AgNW components were fabricated to determine their effects on the material properties. The morphologies of the AA fillers with the AlN-to-AgNO_3_ ratios of 1:2, 1:1, 2:1, and 3:1 are depicted in Appendix A. They show that the AgNW amount on the AlN surface decreases with an increase in the number of AlN particles during the stirring of AlN and AgNO_3_ seeds.

### 3.2. Filler Crystalline Structure

The crystalline structures of the filler and 3D network are shown in Figure 3. The XRD patterns of the raw AlN and AgNW depicted in Figure 3a contain regular peaks, which are consistent with the data reported in previous studies [36,37]. After the growth of AgNWs on the AlN surface, the obtained AA XRD patterns include both the AlN and AgNW peaks, suggesting that the deposited AgNWs retained their crystalline structure. In addition, the 3D network of AA filler (AAN) prepared by sintering AgNWs on the AlN surface produced similar diffraction peaks with the intensities close to those obtained for the AA filler before hot pressing. Therefore, the fabricated 3D network maintained the crystallinity of the filler. To examine the effect of particle sintering on this structure, the crystallite size was calculated by the Scherrer equation [38]:(1)$$D=\;\frac{0.9\lambda }{\beta \cos\theta },$$
where D is the crystallite size, λ is the wavelength of Cu K_α_ radiation, β is the full width at half-maximum of the Bragg peak, and θ is the Bragg angle. The crystallite size of the AA filler before sintering determined by this formula was 26 nm. After the sintering process conducted at a temperature above the T_g_ of AgNWs, the crystallite size of the 3D network was close to 36 nm, suggesting that the sintering caused increase of crystallite size of AgNWs [39].

### 3.3. AgNW Coalescence during Heating

The sintering of the AA fillers is illustrated in Figure 4, which contains the DSC curves recorded for the AlN, AgNW, and AA components at temperatures from 20 to 400 °C. The obtained AA DSC curve exhibits an exothermic peak at 310 °C with an enthalpy of 59.1 J/g, which is consistent with the AgNW data, while the AlN DSC curve does not have an exothermic peak in the same temperature range (Figure 4a). This is because the surface energy of AgNWs should reduce after they are fused to each other [40]. The morphology of the sintered AA particles on the AlN surface was analyzed by TEM (Figure 4b). The obtained image shows the AlN particles connected by sintered the AgNWs to form the 3D AA network.

### 3.4. Morphologies of the 3D Network and Fabricated Composites

The 3D network of the AA fillers in this study were fabricated at different hot pressing times (30 s, 1 min, and 2 min) and AlN-to-AgNW ratios (1:2, 1:1, 2:1, and 3:1). First, the effect of hot pressing time on the morphologies of the 3D network and composites was determined using 1:1 ratio AA filler (Appendix A). The cross-sectional images of the 3D network obtained at hot pressing times of 30 s, 1 min, and 2 min are depicted in Appendix A, while the corresponding composite images obtained after EP impregnation are presented in Appendix A, respectively. These images show that the pore size between the AA fillers decreases with an increase in the hot pressing time during sintering. Moreover, the optimal hot pressing time determined for the prepared composites was 30 s because of the low number of voids depicted in Appendix A. In contrast, more voids were generated by the smaller pores in Appendix A, which were difficult to fill with EP species (1 min and 2 min). In addition, the influence of the Ag wire shape (Appendix A) was confirmed by hot pressing a filler containing Ag nanoparticles on the AlN surface (Appendix A) instead of AgNWs under the same conditions. The resulting 3D network was too fragile to maintain its structure (Appendix A) because the deposited silver nanoparticles (AgNPs) were unable to bridge the fillers. Therefore, the Ag wire shape was critical for preserving the 3D network structure.

The effect of the filler ratio on the composite structure is described in Figure 5. According to the obtained cross-sectional FE–SEM images, the number of voids between fillers in the 3D network increases with increasing AlN content, owing to the decrease in the number of AgNW bridges. After the incorporation of the EP resin, no voids were observed for any of the fabricated composites. The thicknesses of composites prepared at various AlN-to-AgNW ratios are listed in Appendix A. This shows that increasing the AlN fraction in the AA filler increases the composite thickness because of the micro-size of the AlN particles.

### 3.5. Thermal Properties of the Prepared 3D Network Composites

To determine the through-plane thermal conductivities of the fabricated composites, their thermal diffusivities were measured by LFA. The samples in this study were labeled “x−1:1”, where x was the filler weight in milligrams, and “1:1” was the AlN-to-AgNW weight ratio.

Increasing the filler loading from 0 to 400 mg, the thermal conductivities of all composites increased in the same tendency (Figure 6a). Furthermore, the thermal conductivity of the composites was high in the ratio order of 1:1, 1:2, 2:1, and 3:1. The maximum thermal conductivity of 4.49 W m^−1^ K^−1^ was obtained for the 400–1:1 sample with a filler loading of 49.86 vol.%. These results indicate that the heat transfer paths of the 3D network in the EP matrix were most efficiently formed at the 1:1 ratio between the AlN and AgNW components. Meanwhile, too many inefficient thermal transfer paths along the perpendicular direction were present in the network with an AlN-to-AgNW filler ratio of 1:2. In addition, the amount of AgNWs on the AlN surface was not sufficient for bridging the fillers when the AlN-to-AgNW ratio increased from 2:1 to 3:1. Therefore, the heat transfer paths formed at these two ratios were less efficient than those produced at the 1:1 ratio. To characterize the thermal management performance of the synthesized composites quantitatively, the thermal conductivity enhancement (TCE) value was introduced in Figure 6b. Its magnitude was calculated by the following equation [41]:(2)$$\mathrm{TCE}\;\left(\%\right)=\frac{K_{c}-K_{e}}{K_{e}}\times 100,$$
where $K_{c}$ and $K_{e}\;$are the thermal conductivities of the composites and epoxy, respectively.

The highest thermal conductivity of 4.49 W m^−1^ K^−1^ obtained at a filler content of 49.86 vol% represents an enhancement of 1973% with respect to the thermal conductivity of neat EP (0.22 W m^−1^ K^−1^). Therefore, the composites fabricated in this study are highly competitive with the EP composites reported in other works (Figure 6c). Figure 6d shows the results of TGA measurements, which were performed to determine the weight fractions of the EP composite fillers after impregnation and their thermal stabilities at different filler loadings and AlN-to-AgNW ratio of 1:1. The latter were examined by estimating the temperatures at 20% degradation (T_20d_) of these materials. The T_20d_ value increased from 363 to 462 °C (corresponding to an enhancement of 27.28%) with an increase in the filler loading from 0 to 400 mg (the filler weight fractions in the EP composites measured at loadings of 100, 200, 300, and 400 mg were 12.92, 34.38, 43.96, and 63.34 wt%, respectively).

To evaluate the thermal management performance of the fabricated composites, they were applied to central processing units (CPUs) as TIMs (Figure 7). The CPU infrared images obtained during operation and the related temperature–heating time graphs are shown in Figure 7a,b, respectively. A CPU temperature increase rate was observed in the order of 400–1:1, neat EP, and without TIM. Moreover, the maximum CPU temperature of 64.6 °C was achieved when the 400–1:1 composite was used as the TIM, whereas the maximum temperature obtained for neat EP and without TIM were 79.9 and 86.2 °C, respectively. Hence, the thermal management performance of the 3D network composites fabricated in this study was suitable for their use as TIMs in electronic devices.

### 3.6. Electrical Properties of the Prepared 3D Network Composites

To verify the electrical conductivity of the 3D network and composites caused by AgNW addition, the electrical conductivities of the 3D network and composites were measured before and after their impregnation with the EP resin (Figure 8). Compared with the electrical conductivity of the AgNWs (6.3 × 10^5^ S cm^−1^) [20], the electrical conductivities measured at AlN-to-AgNW ratios of 1:2, 1:1, 2:1, and 3:1 decreased to 10.1, 9.3, 9, and 8.7 S cm^−1^, respectively (Figure 8a). This phenomenon resulted from the insulating effect produced by the addition of AlN particles, which impeded the electronic movement between the AgNWs. Therefore, the electrical conductivity of the AA filler decreased with an increase in the AlN content. After the addition of the EP resin, the electrical conductivities were 2 × 10^−10^, 1.8 × 10^−10^, 1.2 × 10^−10^, and 8.3 × 10^−11^ S cm^−1^in order of 1:2, 1:1, 2:1, and 3:1 ratio. These values mean that the composites are insulated by epoxy as calculating into volume resistivity in Figure 8b.

## 4. Conclusions

In this study, a new fabrication method for a 3D filler network was proposed for the first time to prepare thermally conductive EP composites. During the fabrication of the AA filler, AgNWs were grown on the surface of the AlN particles by the polyol method to obtain a chestnut bur-like filler. After the sintering of the AA fillers, the 3D filler network was fabricated by connecting AgNWs on the AlN surface. Subsequently, the EP/AA composites were prepared via the impregnation of the 3D network with the EP resin. The maximum thermal conductivity of 4.49 W m^−1^ K^−1^ was achieved at a filler content of 400 mg (49.86 vol%) represents an enhancement of 1973%. Moreover, the heat transfer paths were most efficiently formed at the same filler ratio. The thermal stability of the composite was enhanced by 27.28% after raising the T_20d_ value from 363 to 462 °C, and its superior thermal management performance as a TIM was confirmed by applying this material to a CPU (operating temperature decrease from 86.2 to 64.6 °C). The electrical conductivity of the produced 3D network was significantly lower than that of the AgNWs, owing to the good insulating properties of AlN. After the infiltration process, the insulating properties of the composites were significantly enhanced after the EP incorporation. Therefore, the highly thermally conductive and insulating EP/AA 3D network composites fabricated in this study can be potentially used as efficient TIMs in electronic devices. Furthermore, the new fabrication method for 3D filler networks described in this study will contribute to developing new TIMs based on such networks.

## Figures and Tables

**Figure 1 polymers-13-00694-f001:**
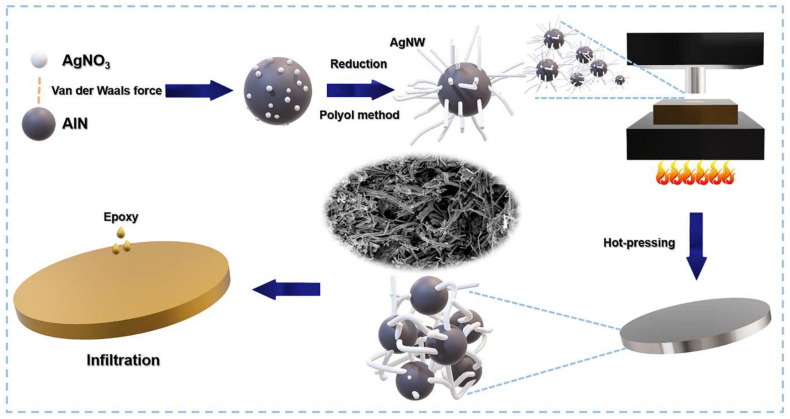
Fabrication process of the epoxy (EP)/ silver nanowire (AgNW)– aluminum nitride (AlN) (AA) 3D network composite.

**Figure 2 polymers-13-00694-f002:**
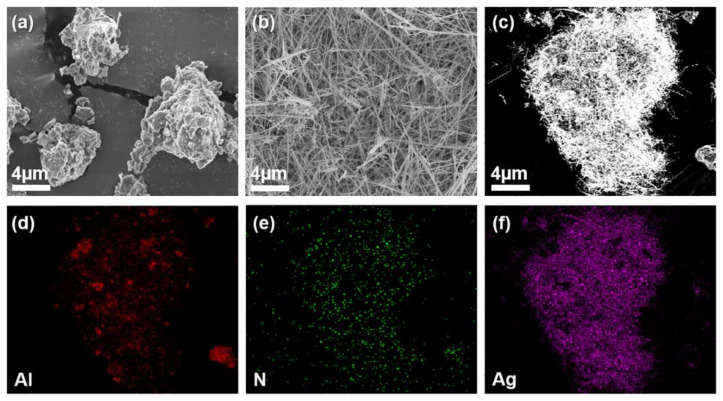
Field–emission scanning electron microscopy (FE–SEM) images of (**a**) raw AlN, (**b**) AgNW, and (**c**) AA. (**d**–**f**) EDS images of AA.

**Figure 3 polymers-13-00694-f003:**
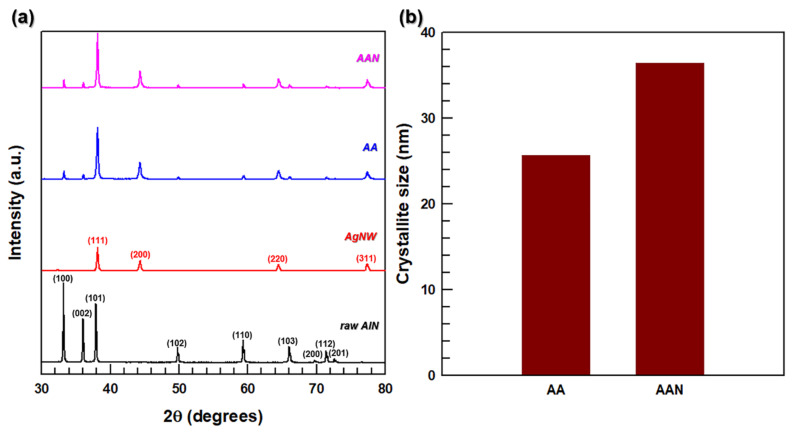
(**a**) X-ray diffraction (XRD) patterns of raw AlN, AgNW, AA, and 3D network of AA filler (AAN). (**b**) crystallite size of AA and AAN.

**Figure 4 polymers-13-00694-f004:**
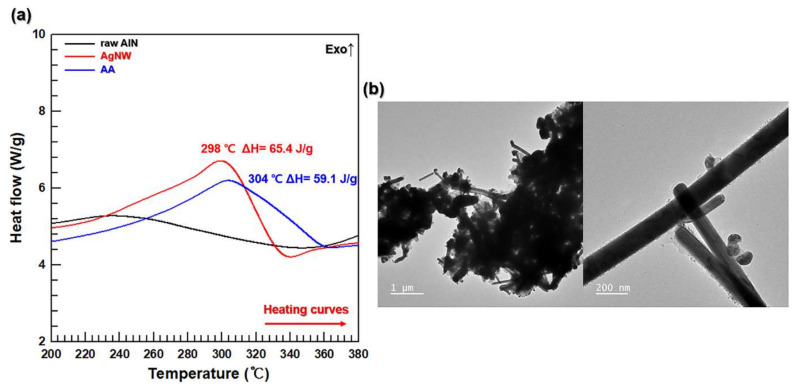
(**a**) Differential scanning calorimetry (DSC) curves of raw AlN, AgNW, and AA. (**b**) FE-TEM images of sintered AA.

**Figure 5 polymers-13-00694-f005:**
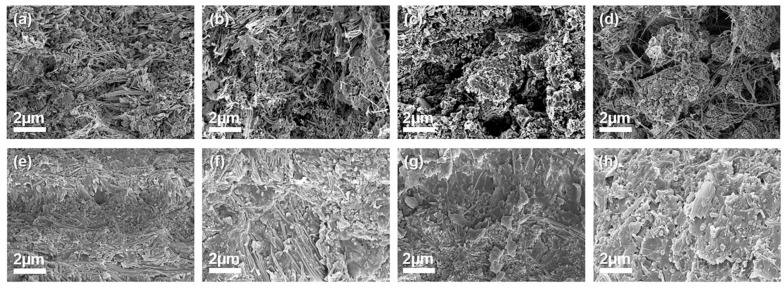
FE-SEM images of cross-sectional surfaces of 3D network ((**a**) 1:2, (**b**) 1:1, (**c**) 2:1, and (**d**) 3:1) and composites ((**e**) 1:2, (**f**) 1:1, (**g**) 2:1, and (**h**) 3:1) along AlN-to AgNW ratio.

**Figure 6 polymers-13-00694-f006:**
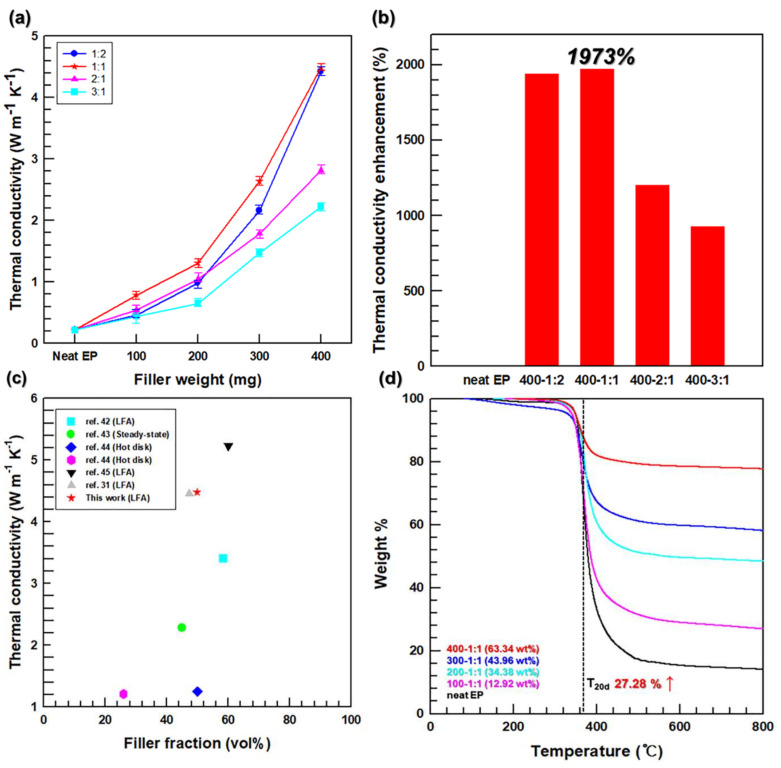
Thermal properties of the composites containing different fillers with (**a**) through-plane thermal conductivity, (**b**) thermal conductivity enhancement, (**c**) through-plane thermal conductivity versus filler fraction of composites in comparison with other thermal conductive composites using AlN [42,43,44,45], and (**d**) thermogravimetric analysis (TGA) curves of EP/AA composites.

**Figure 7 polymers-13-00694-f007:**
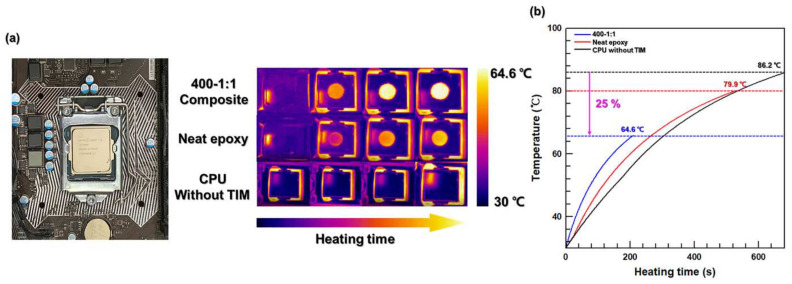
(**a**) Infrared thermograms of the EP/AA composites during the CPU operating and (**b**) corresponding temperature–time curves.

**Figure 8 polymers-13-00694-f008:**
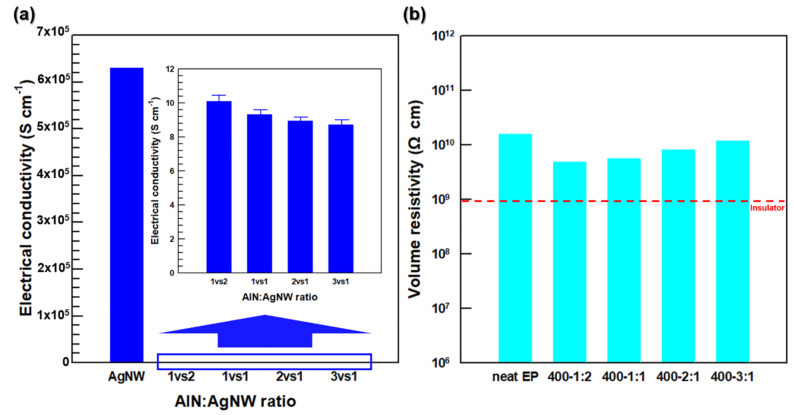
(**a**) electrical conductivity of AgNW and AA 3D network along AlN-to-AgNW ratio. (**b**) volume resistivity of EP/AA composites.

## Data Availability

The data presented in this study are available on request from the corresponding author.

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
