# Peer review of "Highly Thermal Conductive and Electrical Insulating Epoxy Composites with a Three-Dimensional Filler Network by Sintering Silver Nanowires on Aluminum Nitride Surface"

_polymers, 2021, doi:10.3390/polym13050694_

Round 1

Reviewer 1 Report

  1. In the abstract, abbreviations should be avoided. For instance, what is CPU and TIM?
  2. In the abstract, line 15, “Represents” should be “representing”
  3. Line 85: What is EG?
  4. Line 101: What is FTFE?
  5. Lines 114 to 116: Are the Thermal conductivities of the composites measured or calculated?
  6. Line 114: I guess, “converted” should be “calculated”
  7. I suggest to move, section 3.1 to the Experimental section. However, the 3D structure images can be discussed as a result.
  8. Lines 136, 183: Please check the language
  9. Section 3.4: To evaluate the effect of the hot pressing times, it is not clear as to which ratio was used. Should be mentioned clearly in the text.
  10. Line 200: What is AgNPs?
  11. MAJOR CONCERN 1:
    1. One of the most important properties measured by the authors is the Thermal diffusivity of the composite samples with 3D network fillers by the LFA. However, the authors failed to describe the complete step by step procedure used to measure the diffusivity, which is later converted to the thermal conductivity by the given equation. Moreover, the reviewer would like to check if the results shown in Figure 6C, in comparison to the thermal conductivities from literature are all measured in the same way, i.e the thermal diffusivities were measured and then converted or the thermal conudctivities were directly measured. Please clarify.
    2. The Thermal conductivity obtained in Ref 45 seems to be much higher as compared to the present study. So, in what way I this study unique?
  12. The importance of equation 2 is not stated properly. Why is there as suddenly the thermal conductivity of a film and matrix?
  13. I suggest to use the y-axis label as “Electrical Conductivity” in Figure 8a
  14. Line 267: They should refer to the conductivity as “Electrical conductivity” to remove any ambiguity.
  15. MAJOR CONCERN 2: A step-step procedure of measuring the electrical conductivity should be given. I guess, just like the figure 6C given wherein the present values are compared to the one present in literature, should be provided for the electrical conductivities as well.
  16. MAJOR CONCERN 3: The authors have not mentioned as to whether they checked the repeatability of their results. That is how many samples were tested for each of the condition for the thermal diffusivity and the electrical conductivity? If more than one sample was tested then error bars should be added to all the graphs. If only one sample was tested, then how are they claiming their excellent results on the basis of only one experiment? Please clarify.

Author Response

Comments from the editors and reviewers:

The authors thank the reviewer for your valuable suggestions and comments. Please see our responses and rebuttals below.
-Reviewer 1

  1. In the abstract, abbreviations should be avoided. For instance, what is CPU and TIM?

As reviewer mentioned, we added the full words for the abbreviations. The changed sentence below :

From

Moreover, the EP/AA decreased operating temperature of CPU from 86.2 to 64.6 ℃ as a TIM. The thermal stability was enhanced by 27.28 % (99 ℃) and the composites showed insulating after EP infiltration owing to the good insulation properties of AlN and EP.

To

Moreover, the EP/AA decreased operating temperature of central processing unit (CPU) from 86.2 to 64.6 ℃ as a thermal interface material (TIM). The thermal stability was enhanced by 27.28 % (99 ℃) and the composites showed insulating after EP infiltration owing to the good insulation properties of AlN and EP.

  1. In the abstract, line 15, “Represents” should be “representing”

As reviewer mentioned, the word was changed.

  1. Line 85: What is EG?

EG means ethylene glycol, which abbreviation was added in 2.1. Materials.

  1. Line 101: What is FTFE?

FTFE is typo. We fixed the full name “polytetrafluoroethylene (PTFE)”

  1. Lines 114 to 116: Are the Thermal conductivities of the composites measured or calculated?

The thermal conductivites were calculated using equation from measured factors (thermal diffusivity, bulk density, heat capacity)

  1. Line 114: I guess, “converted” should be “calculated”

As reviewer mentioned, the word was changed from “converted” to “calculated.”

  1. I suggest to move, section 3.1 to the Experimental section. However, the 3D structure images can be discussed as a result.

As reviewer mentioned, the part of 3.1 (To achieve excellent heat management performance of the produced composites, a 3D network comprising AlN and AgNWs was fabricated by the polyol method followed by hot pressing.) were moved to experimental section 2.3.

  1. Lines 136, 183: Please check the language

We checked and modified.

  1. Section 3.4: To evaluate the effect of the hot pressing times, it is not clear as to which ratio was used. Should be mentioned clearly in the text.

We added the explanation of ratio. the changed sestence below:

From

First, the effect of hot pressing time on the morphologies of the 3D network and composites was determined by FE–SEM (Fig. S2).

To

First, the effect of hot pressing time on the morphologies of the 3D network and composites was determined using 1:1 ratio AA filler (Fig. S2).

  1. Line 200: What is AgNPs?

We added the full name of AgNPs as silver nanoparticles.

MAJOR CONCERN 1:

  1. One of the most important properties measured by the authors is the Thermal diffusivity of the composite samples with 3D network fillers by the LFA. However, the authors failed to describe the complete step by step procedure used to measure the diffusivity, which is later converted to the thermal conductivity by the given equation. Moreover, the reviewer would like to check if the results shown in Figure 6C, in comparison to the thermal conductivities from literature are all measured in the same way, i.e the thermal diffusivities were measured and then converted or the thermal conudctivities were directly measured. Please clarify.

We added the explanation of procedure used to measure diffusivity in 2.4. Characterization and the label the method used for measuring the thermal conductivity in Fig .6(C). The changed sentence and figure are below :

From

A laser flash analysis (LFA, Netzsch Instruments Co., NanoFlash LFA 467) was used to examine the thermal diffusivities of the composites at room-temperature.

To

A laser flash analysis (LFA, Netzsch Instruments Co., NanoFlash LFA 467) was used to examine the thermal diffusivities and specific heat capacity of the composites with the laser flash pulse directed through a disc (diameter : 12.7mm) specimen at room-temperature.

  1. The Thermal conductivity obtained in Ref 45 seems to be much higher as compared to the present study. So, in what way I this study unique?

Although some studies showed higher thermal conductivity than our study, the 3D network fabrication process using sintering of silver nanowire tried newly in this study, which can contribute to diversify the foward studies.

  1. The importance of equation 2 is not stated properly. Why is there as suddenly the thermal conductivity of a film and matrix?

As reviewer metioned, we changed the explanation.

From

                                                             (2)

where  and are the thermal conductivities of the film and matrix, respectively.

To

                                                             (2)

where  and are the thermal conductivities of the composites and epoxy, respectively.

  1. I suggest to use the y-axis label as “Electrical Conductivity” in Figure 8a

As reviewer mentioned, the label was changed to electrical conductivity. The changed figure is below:

  1. Line 267: They should refer to the conductivity as “Electrical conductivity” to remove any ambiguity.

As reviewer mentioned, all “conductivity” were changed to “electrical conductivity”.

MAJOR CONCERN 2: 16. A step-step procedure of measuring the electrical conductivity should be given. I guess, just like the figure 6C given wherein the present values are compared to the one present in literature, should be provided for the electrical conductivities as well.

Thank you for your careful comments. The electrical conductivities of composites in this study was represented for showing the insulating property and tendency, which come from various ratio of AlN, not superiority of the properties. In these reason, we also represented the volume resistivity and the standard line of the insulator. Thus, we think that the comparison of electrical conductivities with another studies is unnecessary data.

MAJOR CONCERN 3: 17. The authors have not mentioned as to whether they checked the repeatability of their results. That is how many samples were tested for each of the condition for the thermal diffusivity and the electrical conductivity? If more than one sample was tested then error bars should be added to all the graphs. If only one sample was tested, then how are they claiming their excellent results on the basis of only one experiment? Please clarify.

The ten samples used for test. As reviewer mentiond, we added the error bars on the Fig.7(a) and 8(a). The changed figures are below :

Reviewer 2 Report

In this contribution by Lee and Kim, the authors studied how a combination of aluminum nitride, silver nanowires (please refer to Comment 5), and an epoxy resin were combined to create a highly thermally conductive composite. The results are interesting and fit the scope of Polymers. However, there are some issues, which must be addressed before the paper will reach the level of discussion expected from a publication in this journal. Please find the suggestions given below:
1) "The final AA filler product was obtained by filtration and drying in an oven at 70 °C overnight." (Lines 91-92) What membranes/solvents were used for filtration?
2) Acceleration voltage in SEM and TEM may significantly affect the imaging, but the employed values were not specified. 
3) Redundant empty space should be eliminated from the bottom of Pages 3, 7, and 9.
4) SEM micrographs should always be provided at the same magnification to enable comparison between the panels. Please supplement a micrograph of AgNWs with a 4 um scale bar in Fig. 2b. 
5) TEM micrograph should also be given for AgNWs before and after the thermal processing. For now, there is no proof that these are nanowires.
6) A serious concern is the lack of error bars, which casts doubt if these results should be interpreted. Please gather more data, do the error analysis and validate your claims. 
7) Chemical composition of AgNWs before and after sintering should be conducted. Exposure to such high-temperature likely results in oxidation of the material. In this case, silver oxide nanowires would be at the heart of this study, not silver nanowires. 
8) Plots formatting should be uniform to increase the readability of this paper. 

Author Response

Comments from the editors and reviewers:

The authors thank the reviewer for your valuable suggestions and comments. Please see our responses and rebuttals below.

-Reviewer 2

In this contribution by Lee and Kim, the authors studied how a combination of aluminum nitride, silver nanowires (please refer to Comment 5), and an epoxy resin were combined to create a highly thermally conductive composite. The results are interesting and fit the scope of Polymers. However, there are some issues, which must be addressed before the paper will reach the level of discussion expected from a publication in this journal. Please find the suggestions given below:
1) "The final AA filler product was obtained by filtration and drying in an oven at 70 °C overnight." (Lines 91-92) What membranes/solvents were used for filtration?

We added the information of membrane and solvents. The changed sentence is below:

From

The final AA filler product was obtained by filtration and drying in an oven at 70 °C overnight.

To

The final AA filler product was obtained by filtration (nylon membranes, pore size : 0.45 µm, diameter : 47 mm) using ethanol as solvent and drying in an oven at 70 °C overnight.

2) Acceleration voltage in SEM and TEM may significantly affect the imaging, but the employed values were not specified. 

We added the electron high tension voltage of SEM (5kV) and TEM (200kV) in 2.4. Chracterization

3) Redundant empty space should be eliminated from the bottom of Pages 3, 7, and 9.

We eliminated.

4) SEM micrographs should always be provided at the same magnification to enable comparison between the panels. Please supplement a micrograph of AgNWs with a 4 um scale bar in Fig. 2b. 

As reviewer mentioned, we changed silver nanowires image in the Fig.2b. The changed figure is below :

5) TEM micrograph should also be given for AgNWs before and after the thermal processing. For now, there is no proof that these are nanowires.

Thankyou for your careful comments. We already presented the proof of AgNWs on the surface of aluminum nitride using the FE-SEM EDS mode in the Fig.2(d-f). Because the samples were same used for FE-SEM and TEM analysis, we think that EDS data is enough to prove the existence of AgNWs on the surface of Aluminum nitride.

6) A serious concern is the lack of error bars, which casts doubt if these results should be interpreted. Please gather more data, do the error analysis and validate your claims. 

As reviewer mentioned, we gathered more data and added error bar on the Fig. 6(a) and 8(b). The changed figures are below :

7) Chemical composition of AgNWs before and after sintering should be conducted. Exposure to such high-temperature likely results in oxidation of the material. In this case,

silver oxide nanowires would be at the heart of this study, not silver nanowires. 

Thank you for your careful comments. After sintering at 310 ℃, the XRD diffraction was not changed in Fig. 3(a). In other words, the silver oxide peaks [1] are not appeared. Moreover, the silver nanoparticles sintering around 300 ℃ in other study were not oxidized [2]. Thus, silver nanowires seems to be correct to say that it was not oxidized in our procedure.

[1]. Ravichandran, S., Paluri, V., Kumar, G., Loganathan, K., & Kokati Venkata, B. R. (2016). A novel approach for the biosynthesis of silver oxide nanoparticles using aqueous leaf extract of Callistemon lanceolatus (Myrtaceae) and their therapeutic potential. Journal of Experimental Nanoscience11(6), 445-458.

[2]. Chen, C., Xue, Y., Li, Z., Wen, Y., Li, X., Wu, F., ... & Xie, X. (2019). Construction of 3D boron nitride nanosheets/silver networks in epoxy-based composites with high thermal conductivity via in-situ sintering of silver nanoparticles. Chemical Engineering Journal369, 1150-1160.

8) Plots formatting should be uniform to increase the readability of this paper. 

As reviewer mentioned, We unifyed the plots formatting.

We hope that this revision addresses all the reviewer’s comments.

Sincerely
